# Are socioenvironmental factors associated with psychotic symptoms in people with first-episode psychosis? A cross-sectional study of a West London clinical sample

Marc S Tibber,[1] James B Kirkbride,[2] Stanley Mutsatsa,[3] Isobel Harrison,[2] Thomas R E Barnes,[4] Eileen M Joyce,[5] Vyv Huddy[6]

For numbered affiliations see end of article.

**Correspondence to**
Dr Marc S Tibber;
m.tibber@ucl.ac.uk

## ABSTRACT

**Objectives** To determine whether neighbourhood-level socioenvironmental factors including deprivation and inequality predict variance in psychotic symptoms after controlling for individual-level demographics.

**Design** A cross-sectional design was employed.

**Setting** Data were originally collected from secondary care services within the UK boroughs of Ealing, Hammersmith and Fulham, Wandsworth, Kingston, Richmond, Merton, Sutton and Hounslow as part of the West London First-Episode Psychosis study.

**Participants** Complete case analyses were undertaken on 319 participants who met the following inclusion criteria: aged 16 years or over, resident in the study's catchment area, experiencing a first psychotic episode, with fewer than 12 weeks' exposure to antipsychotic medication and sufficient command of English to facilitate assessment.

**Outcome measures** Symptom dimension scores, derived from principal component analyses of the Scale for the Assessment of Positive Symptoms and Scale for the Assessment of Negative Symptoms, were regressed on neighbourhood-level predictors, including population density, income deprivation, income inequality, social fragmentation, social cohesion, ethnic density and ethnic fragmentation, using multilevel regression. While age, gender and socioeconomic status were included as individual-level covariates, data on participant ethnicity were not available.

**Results** Higher income inequality was associated with lower negative symptom scores (coefficient=−1.66, 95% CI −2.86 to −0.46, p<0.01) and higher levels of ethnic segregation were associated with lower positive symptom scores (coefficient=−2.32, 95% CI −4.17 to −0.48, p=0.01) after adjustment for covariates.

**Conclusions** These findings provide further evidence that particular characteristics of the environment may be linked to specific symptom clusters in psychosis. Longitudinal studies are required to begin to tease apart the underlying mechanisms involved as well as the causal direction of such associations.

### Strengths and limitations of this study

▶ This study goes beyond incidence/prevalence-level research to explore the environmental predictors of psychosis at the resolution of symptoms and symptom clusters.

▶ This symptom-level approach has the potential to further our understanding of underlying aetiological mechanisms and is in line with current developments in clinical and research practice.

▶ The study uses a multilevel analysis approach that is perfectly suited to the nested/hierarchical structure of the data.

▶ The main limitation of the study is its cross-sectional design, which precludes inferences about causality.

## BACKGROUND

Epidemiological studies exploring geographical variation in the incidence of psychotic disorders have highlighted various characteristics of environments that are typically associated with an increased risk including high population density,[1] urbanicity,[2] income deprivation,[3] income inequality[1 4 5] and low social cohesion.[6] More recently, researchers have begun to investigate whether such risk factors are associated not with the *incidence* of psychosis, but with the *severity* of psychosis symptoms or symptom cluster scores. Underlying this approach is the notion that exposure to particular types of adverse environments and experiences may be causally linked to the development or exacerbation of specific psychological symptoms.[7] For example, it has been suggested that densely populated, deprived, inner-city areas may be causally associated with the development and expression of positive symptoms, for example, paranoid delusions and hallucinations.[8 9] It has been hypothesised that

this association may be mediated, at least in part, by the increased sense of disempowerment, victimisation and social alienation that (arguably) characterise such neighbourhoods.[10–12]

To date only a handful of studies have used an epidemiological approach to explore the association between socioenvironmental adversity and specific psychosis symptoms or symptom clusters. While these studies have typically reported associations between indices of environmental adversity and some form of positive symptom expression, associations with specific symptom items were not consistent across studies. For example, while Oher et al[13] reported a significant association between neighbourhood-level population density and hallucinations but *not* paranoia in first-episode psychosis (FEP), Wickham et al[14] found an inverse pattern of associations with neighbourhood-level deprivation in the general population: the most deprived neighbourhoods were associated with increased paranoia but *not* hallucinations. In addition, reported associations between socioenvironmental adversity and symptom severity in these studies were not restricted to positive symptoms; associations were also seen with depressive symptoms.[13 14] (See also Johnson et al[15] and Newbury et al.[16])

It is unclear whether this variation—or arguably discrepancy—in findings is due to small effect sizes/chance, or instead, methodological differences between studies. For example, the studies described were undertaken on different population samples (clinical vs non-clinical) at different levels of analysis (within a country at the level of neighbourhoods vs across countries) using distinct indices of socioenvironmental adversity (urbanicity, deprivation and/or inequality) and different symptom measures.[13 14] Nonetheless, the possibility that specific symptom clusters may be linked to defined socioenvironmental risk factors suggests value in moving away from studies of diagnostic incidence towards a focus on the predictors of psychotic symptoms and symptom clusters. By virtue of their greater specificity symptom-level analyses of this kind may be more informative as to the underlying aetiological mechanisms involved, as well as potential targets for intervention.

In a recent study of the symptom dimensions underlying psychotic disorders in a FEP cohort in West London we have shown that symptoms were best characterised by positive, negative and disorganisation syndromes.[17] Here, we examined the extent to which these three symptom dimensions were predicted by neighbourhood-level socioenvironmental factors, including indices of deprivation, inequality and social capital. Given the extant literature in the field[13–16] we predicted that at the syndrome level, indices of socioenvironmental adversity would be associated with higher levels of positive symptoms. Further, given the findings reported by Oher et al[13] in their FEP sample (which most closely matches our own), at an item level of analysis we predicted that socioenvironmental adversity would predict hallucinations, but not paranoia.

## METHODS
### Setting
Cross-sectional data were originally collected as part of the prospective West London First-Episode Psychosis (WLFEP) study.[18 19] Participants presented to secondary care services within the boroughs of Ealing, Hammersmith and Fulham, Wandsworth, Kingston, Richmond, Merton, Sutton and Hounslow, between 1998 and 2006. Ethical approval was obtained from local ethics committees of all boroughs included and written informed consent was obtained (RREC 3006). The study was therefore undertaken in accordance with the ethical standards defined in the 1964 Declaration of Helsinki and subsequent amendments therein.

### Participants
Participants included for participation were aged 16 years or over, experiencing a first psychotic episode, had received fewer than 12 weeks' antipsychotic medication and spoke sufficient English to facilitate assessment. Potential participants were screened using the WHO Psychosis Screen[20] and diagnosed according to Diagnostic and Statistical Manual of Mental Disorders (DSM)-III[21] and DSM-IV criteria[22] by two psychiatric research nurses using the diagnostic module of the Diagnostic Interview for Psychosis.[23] See Tibber et al[17] and Huddy et al[24] for full details.

### Data collection
Information was obtained, with informed consent, from participants' clinical records and clinical interview. Data gathered included basic demographics and symptom scores as well as performance on a number of clinical, cognitive and neuropsychological assessments. All researchers involved in data collection received training to a high standard in the administration of these measures. All data presented were gathered at baseline/initial recruitment to the study.

### Measures
#### Symptom dimension scores
Each participant's symptoms were characterised by three symptom dimension scores, which captured the severity of their positive, negative and disorganisation symptoms. These were derived from a second-order principal component analysis of participants' individual item scores on the Scale for the Assessment of Positive Symptoms (SAPS) and the revised version of the Scale for the Assessment of Negative Symptoms (SANS)[25]; see Tibber et al[17] and Peralta and Cuesta[26] for further details.

#### Individual-level covariates
Basic demographic information including age, gender and place of residence was gathered. Ethnicity data were not recorded. In addition, participants were assigned to one of five socioeconomic categories based on their occupation using the National Statistics Socio-Economic Classification (NS-SEC) system.[27]

## Neighbourhood-level exposures

Each participant's postcode was used to identify their small area neighbourhood, based on the Census Area Statistics (CAS) ward in which they lived at first contact using databases produced by the Office for National Statistics (ONS).[28] For each ward—henceforth referred to as a neighbourhood—a number of indices of urbanicity, deprivation, inequality and social capital were identified; unless specified otherwise, all these data were obtained from the 2001 census.[28]

Following the work of others,[29] population density (measured in people per hectare) was used as a proxy for urbanicity.

Two indices of deprivation were included in the study: (1) ID: the percentage of individuals who were living in a household with an income of less than 60% of the median, and (2) the Index of Multiple Deprivation (IMD), an aggregate measure of deprivation which comprised 37 indicators, obtained from the English Indices of Deprivation (2004). Since these data were available for nested geographical areas smaller than the CAS ward—lower layer super output areas (LSOA)—neighbourhood level indices were calculated as the sum of composite LSOA values, weighted by their population size, as described previously.[1]

In addition, it was also possible to derive indices of *inequality*, that is, the dispersion of deprivation. For both the ID and IMD, a corresponding Gini coefficient was calculated for each neighbourhood (GINI-ID and GINI-IMD), based on the distribution of deprivation across its composite LSOAs, as described previously.[1] A Gini coefficient of zero represents perfect equality, whereas Gini coefficients approaching 1 indicate maximum inequality.

To calculate a Social Fragmentation Index (SFI), a composite of four separate 2001 census measures (Z-transformed and summed) was used, as described previously[30 31]: (1) the percentage of people aged 16 years or over and single; (2) the percentage of households that were single occupancy; (3) the percentage of households that were rented; and (4) the percentage of people who were mobile in the 12 months prior to the census date. As a proxy for social cohesion, voter turnout during the 2002 local elections (percentage of the electorate who cast valid ballots) was obtained for each neighbourhood[32] as described previously: Social Cohesion Index (SCI).[6 13]

Ethnic segregation was estimated using the Index of Dissimilarity (IDS).[6] IDS was calculated at the neighbourhood level, measuring the extent to which white and black/minority/ethnic (BME) populations were segregated across lower output areas (LSOAs) within each neighbourhood (IDS-BME). IDS scores ranged from 0 (no segregation) to 1 (total segregation). In addition, BME ethnic density (DEN-BME) was calculated as the proportion of BME individuals relative to the total population in each neighbourhood.

## Statistical analyses

To determine the neighbourhood-level predictors of psychotic symptoms, symptom dimension scores were regressed on predictors using multilevel linear regression. Prior to regression symptom dimension scores were transformed to minimise skew, using an optimal transformation (square root, cube root, logarithmic or inverse transformations) on the basis of which minimised skew to the greatest extent. The data were then Z-transformed to have a mean of 0 and SD of 1, with extreme outliers (>3Z) discarded.

For all regression analyses a null multilevel model was run first to determine the proportion of variance explained by neighbourhood-level random effects. Basic demographic information (age at assessment, sex and NS-SEC) was then added as potential a priori confounders with neighbourhood-level predictors added subsequently using forward stepwise selection. These were only retained/added to the multivariable model if they significantly improved its fit (p<0.05; likelihood ratio test). NS-SEC and sex were coded as categorical variables, with 'unemployed' and 'male' set as reference levels.

To explore associations at the item-level participants' responses on the persecutory delusions (SAPS-D1) and (global) hallucinations (SAPS-H7) items of the SAPS were also regressed on predictors. Since the distribution of participants' responses on these items could not be normalised through transformation they were recoded as binary variables, with scores of 0–2 coded as absent to mild and scores of 3–5 coded as moderate to severe, and analyses undertaken using multilevel logistic regression.

Regression analyses were undertaken using Stata (V.14; StataCorp, College Station, TX).

## Patient and public involvement

Patients and public were not involved in the design or analysis of this study.

## Data availability

Data from this study are not openly available. The combination of demographic, socioeconomic and geographic information, measured at a fine scale, coupled with the relative low incidence of psychosis would risk compromising participant anonymity.

## RESULTS
### Sample

Unfortunately, information as to the number of potential participants who were evaluated, screened and eventually excluded was not routinely recorded throughout the study. Nonetheless, of the 379 participants for whom data were originally collected, 34 (8.97%) were discarded due to inappropriate residency: no fixed abode (n=5), incomplete postcode information (n=4) or resident beyond the study's catchment area (n=25). Of the remaining 345, complete symptom data were available for 335 participants (97.10%). Following transformation to minimise

**Table 1** Demographic and clinical characteristics of the study sample

| Variable | Level | n (%) | Median | IQR |
|---|---|---|---|---|
| Age | – | – | 24.16 | 20–30.23 |
| Gender | Male | 210 (65.83) | – | – |
| | Female | 109 (34.17) | – | – |
| NS-SEC | Managerial and professional | 17 (5.33) | – | – |
| | Intermediate occupations | 21 (6.58) | – | – |
| | Routine and manual | 50 (15.67) | – | – |
| | Unemployed | 182 (57.05) | – | – |
| | Students | 49 (15.36) | – | – |
| Diagnosis | Schizophrenia | 195 (61.13) | – | – |
| | Schizophreniform disorder | 40 (12.54) | – | – |
| | Brief psychotic disorder | 3 (0.94) | – | – |
| | Delusional disorder | 3 (0.94) | – | – |
| | Schizoaffective disorder | 42 (13.17) | – | – |
| | Bipolar disorder | 20 (6.27) | – | – |
| | Major depression with psychotic features | 10 (3.13) | – | – |
| | Not recorded | 6 (1.88) | – | – |
| DUP | – | – | 12 | 4–44 |
| SAPS total | – | – | 32 | 23–45 |
| SANS total | – | – | 18 | 7–34 |

Statistics provided include the number and percentage of cases (n/%), the median and the IQR. Age refers to age at assessment.
DUP, duration of untreated psychosis (in weeks); NS-SEC, National Statistics Socio-Economic Classification system; SANS, Scales for the Assessment of Negative Symptoms (global scores); SAPS, Scales for the Assessment of Positive Symptoms (global scores).

skew, eight outlying symptom component scores were removed. With respect to basic demographics, seven data points were not coded at the time of data collection: age (n=2), gender (n=2) and occupation (n=3). Further, voter turnout (SCI) was not available for three neighbourhoods due to the existence of a single, unopposed representative. Taken together, these exclusions resulted in a complete core data set of 319 participants, representing 84.17% of the original sample (see table 1). Within this sample the median age was 24.16 years (IQR=20–30.23), 65.83% were male, 57.05% were unemployed and 75.55% experienced non-affective psychosis. The median SAPS and SANS total global scores were 32 (IQR=23–45) and 18 (IQR=7–34), respectively. Table 1 also shows the duration of untreated psychosis (DUP, measured in weeks) in order to facilitate comparison with other studies (median=12; IQR=4–44). While DUP was not included as an a priori confounder (since there is previous evidence this is not associated with neighbourhood-level factors in London[33]), rerunning our primary analyses with its inclusion did not impact on the findings (data available from authors). Participants retained in the analyses did not differ from the original data set on any of the individual or neighbourhood-level variables (data available upon request).

### Neighbourhood data
Data included were distributed across 113 neighbourhoods from 14 boroughs. The median number of LSOAs per neighbourhood was 7 (IQR=6–8) and the median number of participants per neighbourhood was 2 (IQR=1–4). In order to characterise the neighbourhoods further the pattern of associations between key measures (see online supplementary table 1) was explored. Spearman's correlations (see online supplementary table 2) indicated that, at a corrected alpha of 0.002 (corrected for 21 comparisons), the more deprived neighbourhoods (high IMD) were significantly more densely populated (p<0.001), less unequal (p<0.001), more socially fragmented (p<0.001), less socially cohesive (p<0.001) and characterised by higher ethnic segregation (p<0.01) and a higher density of ethnic minorities (p<0.001). Further, more unequal neighbourhoods were significantly less densely populated (p<0.001), less deprived (p<0.001), less socially fragmented (p<0.001), more socially cohesive (p<0.001) and had a lower density of ethnic minorities (p<0.001).

### Multilevel modelling of symptom components and symptom scores
#### Fixed effects
Analyses undertaken while controlling for age, gender and social class ($\chi^2_{(7)}$=24.25, p=0.001) indicated that inequality (GINI-ID) was significantly associated with negative symptoms (table 2 and online supplementary table 3): higher inequality was associated with less severe symptoms (coefficient=−1.54, 95% CI −2.76 to −0.33, p=0.01). This effect

**Table 2** Multilevel modelling of symptom components (summary)

| Predictor | Level | Negative symptoms | | Positive symptoms | | Disorganisation symptoms | |
|---|---|---|---|---|---|---|---|
| | | Fixed part of the model | | Fixed part of the model | | Fixed part of the model | |
| | | Coefficient (95% CI) | Wald P value | Coefficient (95% CI) | Wald P value | Coefficient (95% CI) | Wald P value |
| Age | | 0.01 (−0.01 to 0.02) | 0.37 | −0.01 (−0.02 to 0) | 0.12 | −0.01 (−0.02 to 0.01) | 0.33 |
| Gender | Female | −0.36 (−0.59 to -0.12) | **<0.01** | 0.06 (−0.18 to 0.3) | 0.63 | −0.23 (−0.47 to 0.01) | 0.06 |
| NS-SEC | Managerial | −0.49 (−0.98 to 0) | 0.05 | 0.03 (−0.47 to 0.53) | 0.91 | −0.22 (−0.73 to 0.29) | 0.39 |
| | Intermediate | −0.23 (−0.67 to 0.22) | 0.32 | 0.49 (0.05 to 0.94) | **0.03** | −0.08 (−0.54 to 0.37) | 0.72 |
| | Routine | −0.35 (−0.65 to -0.04) | **0.03** | 0.16 (−0.15 to 0.47) | 0.32 | −0.26 (−0.57 to 0.05) | 0.1 |
| | Student | 0.09 (−0.23 to 0.4) | 0.59 | 0.07 (−0.25 to 0.39) | 0.67 | −0.21 (−0.53 to 0.12) | 0.21 |
| | | | | | | | |
| *Pop Den* | | – | – | – | – | – | – |
| *ID* | | – | – | – | – | – | |
| *IMD* | | – | – | – | – | – | – |
| *GINI-ID* | | −1.66 (−2.86 to −0.46) | **<0.01** | – | – | – | – |
| *GINI-IMD* | | – | – | – | – | – | – |
| *SFI* | | – | – | – | – | – | – |
| *SCI* | | – | – | – | – | – | – |
| *IDS-BME* | | – | – | −2.32 (−4.17 to −0.48) | **0.01** | – | – |
| *DEN-BME* | | – | – | – | – | – | – |

Individual symptom dimensions derived from second-order principal component analysis were regressed on predictor variables using multilevel regression analyses. Data reported are from multivariate models that were run while controlling for basic demographic information (age, gender and NS-SEC) and other symptoms (eg, negative and disorganisation symptoms for the positive symptoms). Neighbourhood-level variables are in italics. Significant variables are in bold.
BME, black/minority/ethnic; DEN-BME, BME ethnic density (people per hectare); GINI, Gini coefficient; GINI-ID, Gini coefficient based on index of deprivation; GINI-IMD, Gini coefficient based on index of multiple deprivation; ID, index of deprivation; IDS, index of dissimilarity; IDS-BME, index of dissimilarity for BME versus white populations; IMD, index of multiple deprivation; NS-SEC, National Statistics Socio-Economic Classification system; Pop Den, population density; SCI, social cohesion index; SFI, social fragmentation index.

persisted after further control for other (positive and disorganisation) symptoms (coefficient=−1.66, 95% CI −2.86 to −0.46, p<0.01) as well as absolute deprivation, that is, ID or IMD (coefficient=−2.06, 95% CI −3.32 to −0.8, p<0.01; coefficient=−2.06, 95% CI −3.32 to −0.8, p<0.01). Further, ethnic segregation (IDS-BME) was associated with positive symptoms after controlling for basic demographics: higher segregation of ethnic minorities was associated with less severe positive symptoms (coefficient=−2.36, 95% CI −4.2 to −0.52, p=0.01; table 2 and online supplementary table 4). This effect also persisted after controlling for other symptoms (coefficient=−2.32, 95% CI −4.17 to −0.48, p=0.01) as well as absolute deprivation, that is, ID or IMD (coefficient=−2.2, 95% CI −4.09 to −0.32, p=0.02; coefficient=−2.17, 95% CI −4.07 to −0.28, p=0.03). However, no significant neighbourhood-level predictors were found for disorganisation symptoms (table 2 and online supplementary table 5). Nor did any neighbourhood-level predictor predict significant variance in paranoid delusions or global hallucinations (symptom-level analyses; online supplementary table 6).

### Random effects
Neighbourhood-level random effects did not explain significant levels of variance in symptom dimension or individual symptom scores, either before or after controlling for demographic variables: negative symptoms (<0.001%), positive symptoms (<0.001%), disorganisation symptoms (1.45%), paranoid delusions (5.35%) or global hallucinations (<0.001%) (all p>0.05).

### DISCUSSION
Fixed effects analyses revealed that each symptom dimension (positive, negative and disorganisation) was associated with a distinct pattern of neighbourhood-level risk factors; thus, higher levels of inequality were associated with lower negative symptoms, and greater ethnic segregation was associated with lower positive symptoms. However, contrary to our primary hypotheses, there was no evidence to suggest that the positive symptoms of psychosis were elevated in areas scoring highly on classical indicators of environmental adversity or socioeconomic deprivation (eg, population density, ID or inequality), either at the level of the symptom cluster, or with respect to persecutory delusions or hallucinations specifically.

Nonetheless, the finding that participants living in neighbourhoods characterised by highly segregated BME communities exhibited *less* severe positive symptoms is interesting, since the IDS speaks to the richness of the

social environment. Thus, while a high IDS indicates high segregation *between* populations, it implies reduced fragmentation *within* a given ethnic group, that is, a high potential for bonding social capital,[34 35] with potential protective effects against the positive symptoms of psychosis.[36] This is consistent with a previous study undertaken in South East London, which reported a lower incidence of psychosis in areas characterised by higher (BME) ethnic segregation,[6] as well as research showing that the risk of psychosis diminishes as the proportion of one's own ethnic group increases within a neighbourhood.[36–39]

The finding that participants living in neighbourhoods characterised by higher inequality (GINI-ID) exhibited less severe negative symptoms is inconsistent with the income inequality hypothesis,[40] which posits that highly unequal neighbourhoods are characterised by poorer health outcomes. However, the empirical findings on the association between income inequality and mental health are not clear cut. A recent qualitative synthesis identified 27 relevant studies that explored this issue,[41] nine of which met inclusion criteria for a meta-analysis; of these, only one found a positive association between higher inequality and poorer mental health, while six reported mixed results, and two found no significant effects. The authors concluded that the extant literature in the field is characterised by small effect sizes and a high level of heterogeneity in findings. See also Wilkinson and Pickett[42] and Tibber.[43]

The association between inequality and psychosis—rather than mental health in general—is also not clear cut. For example, while no association was seen between income inequality and psychotic *symptoms* in a study of FEP undertaken in South East London and Nottinghamshire,[13] links between inequality and psychosis *incidence* or *prevalence* have been shown using sample sizes comparable to our own.[1 4 44] It is not inconceivable, however, that inequality and/or deprivation might be characterised by relatively independent (or even inverted) patterns of association with psychosis incidence and psychotic symptom severity. For example, one might imagine how in a wealthy neighbourhood, in which health and social services are well resourced (for detection as well as treatment), the recorded incidence of psychosis might be elevated but the severity of symptoms in those individuals reduced.

Another potential explanation for the unexpected finding of an association between higher inequality and lower negative symptom scores resides in the sociopolitical geography of the West London region examined. According to the social capital hypothesis[45] areas that are relatively more unequal, that is, defined by a higher Gini coefficient, tend to have worse health outcomes because they are characterised by lower levels of interpersonal trust and a lack of a sense of shared community (ie, low social capital and high social fragmentation). Thus, poor social cohesion/social capital is thought to mediate the negative effects of inequality on health.[46] However, within the WLFEP data set the inverse was true (table 2): more

unequal neighbourhoods were in fact *more* socially cohesive, characterised by extremes of affluence (rather than deprivation) and a relative abundance of social capital. Although a direct association was not seen between social cohesion and psychotic symptoms in our study, it is possible that more sensitive indices of social capital, or measures that tap into different facets of this complex construct,[47] might uncover such an association.[48]

Finally, the reported association between higher inequality and lower negative symptoms might be explained by the mixed neighbourhood hypothesis (MNH).[49–51] According to the MNH, the mixing of individuals from different socioeconomic backgrounds within areas of high inequality may be protective against cultures of crime, substance use, joblessness and a lack of social opportunity that can become endemic to areas of homogenous deprivation. Further, the wealthy may also bring higher investment in local infrastructure and resources. Arguably, however, despite considerable political interest in the area, particularly in the USA, reviews of the literature have typically failed to find convincing evidence for the MNH[17 41] or the benefits of mixed housing schemes that are linked to the theory.[52]

With respect to the limitations of the study, there are several. First, its cross-sectional design precludes inferences about causality. Second, the use of national census data from a single time point (2001) in conjunction with participant data collected across a broad temporal window (1998–2006) is likely to have increased measurement error, and therefore, potentially reduced the chance of finding a significant effect (inflation of type II error). Nonetheless, this could not be avoided since national census data are only collected every 10 years.

With respect to the variables available for analysis, while ethnicity was controlled for at the ward level, individual-level ethnicity was not collected. Consequently, individual-level ethnicity may have confounded some of the effects of ward-level indices, particularly those relating directly to ethnicity, for example, the effect of ethnic segregation on positive symptoms. Thus, there is ample evidence that the incidence of psychotic disorders is elevated in migrant and minority ethnic populations, an effect that seems to persist even after controlling for individual-level socioeconomic status.[53] Further, psychotic symptoms may also be elevated in individuals from these populations.[54] However, there is evidence to suggest that ethnicity does not predict differences in psychosis symptom dimension scores; see Oher *et al*[13] for example.

The use of voter turnout as a proxy for social cohesion has some limitations. While there is evidence to suggest that voter turnout does correlate with self-reported interpersonal and societal trust, for example,[55] social cohesion is a complex and multifaceted construct, and as such, is unlikely to be fully captured by a sole crude measure, that is, it may have limited content validity; see Orford.[35] Second, voter turnout, by definition, cannot represent individuals who are denied access to the electoral role, for example, individuals under the voting age (<16 years),

non-citizen migrants, refugees and asylum seekers. This is particularly relevant to the field of psychosis research, since migrant groups have an elevated risk of experiencing psychotic symptoms; see Siegler[56] for a review. Despite these limitations, however, voter turnout has been highlighted as a headline measure of civic and political participation by the Organisation for Economic Co-operation and Development (OECD), has been included in the ONS' National Well-being Wheel of Measures and, further, has been put forward as a sustainable development indicator by the Eurostat/OECD/United Nations Economic Commission for Europe (UNECE) Task Force on Measuring Sustainable Development; see Bryan and Jenkins[57] for further details.

Finally, since our sample was characterised by relatively few participants per neighbourhood we were likely underpowered to detect random effects, particularly in view of the fact that estimates of random effects are comparatively unreliable and biased towards underestimation.[58] Nonetheless, consistent with our a priori aims, neighbourhood covariates (ie, fixed effects) were still modelled and significant associations found.

In conclusion, the findings reported contribute to a growing body of evidence that highlights the importance of social and socioeconomic factors in the expression of psychosis symptoms. Further, they suggest that at least some of the association between psychosis and the environment may be operating at the level of symptoms, with specific environmental predictors linked to specific symptoms or symptom clusters. Arguably, this lends support to current calls in both clinical[59] and research[60] practice for dimension-based models of psychosis,[61–63] a debate that is also echoed more broadly in the field of general psychopathology.[63 64] Finally, the findings also highlight the need for longitudinal research that can begin to elucidate the underlying mechanisms that link the environment to symptom expression, in addition to establishing the direction of causality.

**Author affiliations**
¹Department of Clinical, Educational and Health Psychology, University College London, London, UK
²Division of Psychiatry, University College London, London, UK
³School of Health Sciences, City, University of London, London, UK
⁴Division of Psychiatry, Imperial College London, London, UK
⁵UCL Queen Square Institute of Neurology, University College London, London, UK
⁶Clinical Psychology Unit, Department of Psychology, University of Sheffield, Sheffield, UK

**Contributors** EJ and TREB designed and oversaw the West London First-Episode Psychosis (WLFEP) study. SM, IH and VH tested the participants. VH and MST designed the current study. MST undertook all analyses and wrote the manuscript. JK provided advice and support on statistical analyses. All authors commented on and contributed to the final version of the manuscript.

**Funding** The West London First-Episode Psychosis study was funded by a Wellcome Trust Programme grant to EJ, TREB, Maria Ron and Gareth Barker (Grant No 064607/Z/01/Z). JK was supported by a Sir Henry Dale Fellowship, jointly funded by the Wellcome Trust and the Royal Society (Grant No 101272/Z/13/Z). EJ and JK were supported by the National Institute for Health Research University College London Hospitals Biomedical Research Centre.

**Competing interests** None declared.

**Patient consent for publication** Not required.

**Ethics approval** Ethical approval was obtained from local ethics committees of all boroughs included (RREC 3006).

**Provenance and peer review** Not commissioned; externally peer reviewed.

**Data availability statement** No data are available.

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
