## [Reviewer comments · BMJ Open]

ARTICLE DETAILS

TITLE (PROVISIONAL)	Are socio-environmental factors associated with psychotic symptoms in people with first-episode psychosis?: a cross-sectional study of a west London clinical sample
AUTHORS	Tibber, Marc; Kirkbride, James; Mutsatsa, Stanley; Harrison, Isobel; Barnes, Thomas; Joyce, Eileen; Huddy, Vyv

VERSION 1 – REVIEW

REVIEWER	Catherine Shiu-yin Chong Kwai Chung Hospital, Hospital Authority, Hong Kong Special Administrative Region
REVIEW RETURNED	20-Apr-2019

GENERAL COMMENTS	This is a very interesting piece of article trying to explore the effects of neighbourhood socioeconomic characteristics on psychotic symptoms. However, one major flaw of this study is that, the key dependent variables were inferred from census data in year 2001 for all the subjects recruited between 1998 and 2006. The use of voter turnout rate as a proxy of social cohesion seems inappropriate due to the multiple factors involved. Also, other factors which may contribute to psychotic symptom severity, e.g. duration of untreated psychosis, mode of onset, etc, were not included in the analysis. References quoted were appropriate. Limitations should also discussed the issues raised above if they were not amenable.
--

REVIEWER	Luyken Stouten Parnassia Center for Early Psychosis, The Hague, The Netherlands
REVIEW RETURNED	06-May-2019

GENERAL COMMENTS	Clear paper on an interesting and relevant issue. No specific comments.
---

REVIEWER	Vincent Russell Department of Psychiatry, Royal College of Surgeons in Ireland
REVIEW RETURNED	29-May-2019

GENERAL COMMENTS	The authors explore the association between neighbourhood level socio-environmental factors and variance in psychotic symptoms based on secondary data from a sample of 319 patients from the West London First-Episode Psychosis Study. They hypothesised, based on previous literature, that after controlling for individual-level demographics socioenvironmental adversity would be
--

	associated with higher levels of positive psychotic symptoms and specifically, hallucinations rather than paranoid symptoms. Demographic data other than ethnicity was collected from participants and symptom measures included the SAPS & SANS while neighbourhood level exposures was identified from participants post code and various indices of socioenvironmental were employed. The statistical analysis involved multi-level linear regression. The findings were that, contrary to the hypotheses, elevated scores of positive symptoms were not associated with living in areas of adversity/deprivation. The authors highlight the added finding that participants living in highly segregated BME communities exhibited less severe psychotic symptoms. They acknowledge some of the limitations of the study and interpret and discuss the findings with reference to previous literature. There are several issues that weaken the findings as reported and consequently the interpretation and conclusions. Specific Comments: Abstract: The abstract omits the key fact that ethnicity was not controlled for as an individual level demographic Methods: Page 8 para 2 states that the method used to adjust for the fact that the geographical areas being smaller than the CAS ward were described in Barnes et al 2000 paper but this is not evident from reading this paper as referenced. There is no detail on the specific diagnoses of the participants which limits comparison with other studies Results: Table 1 provides only the number of participants in "affective" and "non-affective" diagnostic groupings - insufficient diagnostic detail. Discussion: The claimed finding in Para 2 relating to participants in highly segregated BME communities as exhibiting less severe psychotic symptoms is indeed interesting but cannot be justified in that the ethnicity of participants was not recorded. So while they may reside in BME areas, whether or not they are members of a BME community is highly relevant and this is unknown. The conclusions, in large part, do not follow from the results. In the concluding paragraph the authors state that the findings suggest that associations between psychosis and the environment may be operating at the level of symptoms rather than at the level of incidence/prevalence. However, this was not the focus of study and no comparative data in incidence/prevalence of psychosis was provided to justify this statement.
--	---

VERSION 1 – AUTHOR RESPONSE

Reviewer 1's comments:

- One major flaw of this study is that, the key dependent variables were inferred from census data in year 2001 for all the subjects recruited between 1998 and 2006.

Unfortunately this was unavoidable since UK national census data are only collected every ten years. Thus, whilst census data were also available for 1991 and 2011, these would clearly be less appropriate for use given the years across which our (participant) data were collected. Nonetheless, it is true that the use of census data that do not perfectly temporally align with the participant data is not ideal, and will have added measurement error, and hence potentially reduced the chance of finding a significant effect (inflation of type II error). We have added the following sentence, however, to highlight this limitation:

“Second, the use of national census data from a single time-point (2001) in conjunction with participant data collected across a broad temporal window (1998-2006) is likely to have increased measurement error, and therefore, potentially reduced the chance of finding a significant effect (inflation of type II error). Nonetheless, this could not be avoided since national census data are only collected every ten years.”

- The use of voter turnout rate as a proxy of social cohesion seems inappropriate due to the multiple factors involved.

We agree that voter turnout is a crude and reductionistic measure of what is a complex, multi-dimensional construct, i.e. social cohesion and its related construct of social capital. Nonetheless, there is evidence to suggest that it does correlate with self-reported interpersonal and societal trust, and has it has been recommended by a number of important organisations (e.g. the OECD) as a key measure of civic and political participation. Nonetheless, we state its limitations within the Discussion section of the manuscript:

“Although a direct association was not seen between social cohesion and psychotic symptoms in our study, it is possible that more sensitive indices of social capital, or measures that tap into different facets of this complex construct [47], might uncover such an association [48].”

Nonetheless, to make this limitation/s more explicit / emphasised, we have now added the following paragraph to the limitations section:

“The use of voter turnout as a proxy for social cohesion has some limitations. Whilst there is evidence to suggest that voter turnout does correlate with self-reported interpersonal and societal trust, e.g. [56], social cohesion is a complex and multi-faceted construct, and as such, is unlikely to be fully captured by a sole crude measure, i.e. it may have limited content validity; see [36]. Second, voter turnout, by definition, cannot represent individuals who are denied access to the electoral role, for example, individuals under the voting age (<16years), non-citizen migrants, refugees and asylum seekers. This is particularly relevant to the field of psychosis research, since migrant groups have an elevated risk of experiencing psychotic symptoms; see [57] for a review. Despite these limitations however, voter turnout has been highlighted as a headline measure of civic and political participation by the Organisation for Economic Co-operation and Development (OECD), has been included in the Office for National Statistics' National Well-being Wheel of Measures, and further, has been put forward as a sustainable development indicator by the Eurostat / OECD / UNECE Task Force on Measuring Sustainable Development; see [58] for further details.”

-Also, other factors which may contribute to psychotic symptom severity, e.g. duration of untreated psychosis, mode of onset, etc, were not included in the analysis.

There may inevitably be many factors that contribute to psychotic symptom severity, not all of which were –or indeed could be- measured in the original study. Nonetheless, interestingly, information on the duration of untreated psychosis (DUP) was actually included in the original data-set. However, this was not hypothesised to be an a priori confounder as there is previous evidence that DUP is not associated with neighbourhood factors. Nonetheless, in the updated version of the manuscript, we have now included data on the DUP to table 1 in order to facilitate comparisons with other studies. However, we make clear that it was not included as a predictor in subsequent (primary) analyses, although doing so does not impact on the findings:

“Table 1 also shows the duration of untreated psychosis (DUP, measured in weeks) in order to facilitate comparison with other studies (median=12; IQR=4-44). Whilst DUP was not included as an a priori confounder (since there is previous evidence this is not associated with neighbourhood level factors in London [34]), re-running our primary analyses with its inclusion did not impact upon the findings (data available from authors).”

- Limitations should also discussed the issues raised above if they were not amenable.

As discussed above, relevant limitations are now explicitly acknowledged in-depth in the limitations section of the manuscript.

Reviewer 2’s comments:

- No comments.

Reviewer 3’s comments:

- Abstract: The abstract omits the key fact that ethnicity was not controlled for as an individual level demographic

This limitation has now been made explicit in the abstract. We have now also included a discussion of this limitation in the limitations section of the Discussion:

“With respect to the variables available for analysis, whilst ethnicity was controlled for at the ward-level, individual-level ethnicity was not collected. Consequently, individual-level ethnicity may have confounded some of the effects of ward-level indices, particularly those relating directly to ethnicity, e.g. the effect of ethnic segregation on positive symptoms. Thus, there is ample evidence that the incidence of psychotic disorders is elevated in migrant and minority ethnic populations, an effect that seems to persist even after controlling for individual-level SES [54]. Further, psychotic symptoms may also be elevated in individuals from these populations [55]. However, there is evidence to suggest that ethnicity does not predict differences in psychosis symptom dimension scores; see [13] for example.”

- Methods: Page 8 para 2 states that the method used to adjust for the fact that the geographical areas being smaller than the CAS ward were described in Barnes et al 2000 paper but this is not evident from reading this paper as referenced.

I am confused by this point. The relevant reference indicated on Page 8 paragraph 2 (as noted by the reviewer) is in fact reference [1], which is the following:

Kirkbride JB, Jones PB, Ullrich S, et al. Social deprivation, inequality, and the neighborhood-level incidence of psychotic syndromes in East London. *Schizophr Bull* 2014;40:169–80.

...not the reference by Barnes et al. (2000) alluded to by the reviewer, which is in fact reference [19] in our manuscript. The methodology we reference is definitely described in Kirkbride et al. (2014) as we report; I have re-checked. Thus, please see bottom of page 171 therein.

- There is no detail on the specific diagnoses of the participants which limits comparison with other studies.

We have now rectified this so information about the number and percentage of participants falling into each diagnostic category (Schizophrenia, Schizophreniform disorder, Brief psychotic disorder, Delusional disorder, Schizoaffective disorder, Bipolar disorder, Major depression with psychotic features) is included. Please see Table 1.

- Results: Table 1 provides only the number of participants in "affective" and "non-affective" diagnostic groupings - insufficient diagnostic detail.

Please see response to point above.

- Discussion: The claimed finding in Para 2 relating to participants in highly segregated BME communities as exhibiting less severe psychotic symptoms is indeed interesting but cannot be justified in that the ethnicity of participants was not recorded. So while they may reside in BME areas, whether or not they are members of a BME community is highly relevant and this is unknown.

As described above, this limitation has now been acknowledged and discussed in the limitations section of the discussion.

- The conclusions, in large part, do not follow from the results. In the concluding paragraph the authors state that the findings suggest that associations between psychosis and the environment may be operating at the level of symptoms rather than at the level of incidence/prevalence. However, this was not the focus of study and no comparative data in incidence/prevalence of psychosis was provided to justify this statement.

This is a good point. We were basing this conclusion on the fact that no single predictor was associated with all symptom dimensions, but instead, a different pattern of predictors was associated with each symptom dimensions. Nonetheless, in the absence of parallel data psychosis incidence / prevalence as well as symptom severity, it is perhaps fairer to say that the findings are consistent with at least some of the association between psychosis and the environment operating at the level of symptoms. The conclusion section has therefore now been edited / re-written in order to reflect this more modest assessment of the paper's implications.

- Change Supplementary Figures 3-5 citation into Supplementary Tables 3-5 to avoid further confusion.

We have now corrected this.